# Tellurite Glasses from the 70TeO_2_-5XO-10P_2_O_5_-10ZnO-5PbF_2_(X= Pb, Bi, Ti) System Doped Erbium Ions—The Influence of Erbium on the Structure and Physical Properties

**DOI:** 10.3390/ijms24043556

**Published:** 2023-02-10

**Authors:** Katarzyna Pach-Zawada, Magdalena Leśniak, Katarzyna Filipecka-Szymczyk, Edmund Golis, Maciej Sitarz, Dominik Dorosz, Jacek Filipecki

**Affiliations:** 1Department of Experimental and Applied Physics, Faculty of Science and Technology, Jan Dlugosz University in Czestochowa, Al. Armii Krajowej 13/15, 42-200 Czestochowa, Poland; 2Faculty of Material Science and Ceramics, AGH University of Science and Technology, Al. Mickiewicza 30, 30-059 Cracow, Poland

**Keywords:** tellurite glasses, defects, structural properties, magneto-optical properties

## Abstract

In this article, we present research on the influence of erbium ions on the structure and magneto-optical properties of 70TeO_2_-5XO-10P_2_O_5_-10ZnO-5PbF_2_ (X = Pb, Bi, Ti) tellurite glass systems. Structural changes occurring in the glasses during doping with erbium ions were investigated using positron annihilation lifetime spectroscopy (PALS) and Raman spectroscopy. The X-ray diffraction (XRD) method was used to confirm the amorphous structure of the investigated samples. Based on the Faraday effect measurements and calculated values of Verdet constant, the magneto-optical properties of the glasses were determined.

## 1. Introduction

Tellurite glasses with heavy metal oxides have found increasing and wide applications in modern science and technology due to their interesting physical and chemical properties. Especially, they exhibit high refractive index (above 2), good mechanical properties, good corrosion resistance, and high thermal stability. They have low crystallization ability and their transformation temperature (T_g_) decreases with an increase in Te content [1]. The TeO_2_-based glasse sare interesting for optoelectronics, because the tellurite’s infrared transmittance range reaches 7 µm [1,2]. As compared with silicate, tellurite glasses are characterized by low phonon energy (750 cm^−1^), which increases the radiant transitions probability and energy level lifetimes of lanthanide ions [2,3]. Tellurite glasse sare promising materials in applications such as active optical fibers, erasable optical recording media, optical switching devices, laser hosts, or Raman amplification. By changing the matrix components of the glass matrix or by doping the glasses with rare earth elements, it is possible to change the optical properties [3,4]. Glasses with strong magneto-optical properties due to doping with paramagnetic rare-earth ions are attractive for applications in optoelectronics [4]. Due to their magnetic physicochemical properties, they create the possibility of applications in optoelectronics, including magnetic field sensors or optical insulators in laser systems when coupling semiconductor lasers with optical fibers to eliminate light reflected from the face of the optical fiber. As a consequence, incorporation of the active lanthanide ions (Ln^3+^) with partially filled 4f shell into the glass matrix results in new specific properties of the glass system. In order to avoid glass crystallization, it is important to properly select the concentrations of impurities, which significantly affects the ion radiation of the lanthanides present in the glass matrix [5,6,7]. The glass structure is characterized by the occurrence of areas with different degrees of order present in the form of coordinating polyhedra creating glass lattice and having their crystalline equivalents. There are also voids (connection areas) in the glass structure where glass modifying ions may be present [7,8].

In order to describe the real structure of glasses, in addition to conventional methods such as X-ray, electron or neutron diffraction, diffraction of high-energy synchrotron radiation, Raman, and IR spectroscopy, it is necessary to use precursor experimental methods that are especially sensitive to free volumes in the glass structure. One of the innovative methods is positron annihilation, namely the positron annihilation lifetime spectroscopy PALS [9]. Annihilation converts matter into energy, especially the mutual conversion of a particle and an antiparticle into electromagnetic radiation. It is evident that, as in any physical process, all conservation laws are met here, i.e., the laws of conservation of energy, momentum, angular momentum, charge, and parity [10]. Thus, when annihilating positron-electron pairs, the mass of these particles is converted into the equivalent energy of gamma photons. Hence, the study of photons resulting from the annihilation process provides information about the state of the annihilating positron-electron pair. If a positron collides with a free electron, then these particles are annihilated with simultaneous emission of an even (2γ) or odd (3γ) number of gamma quanta. In addition to free annihilation, in some materials, a positron can annihilate with an electron in a bound state, creating a hydrogen-like system called a positronium (Ps) [10,11]. Due to the fact that we have different settings of spins of the e + e^−^ pair, we distinguish two types of Ps: para-positronium (p-Ps) with anti-parallel spin arrangement (2γ annihilation) and ortho-positronium (o-Ps) with parallel spin alignment (annihilation on 3γ). Measurements of the positron annihilation lifetime spectra allow for a distinction of three components, i.e., τ_1_, τ_2_, and τ_3_ lifetimes and their corresponding intensities I_1_, I_2_, and I_3_ (I_1_ + I_2_ + I_3_ = 100%). The component τ_1_ is responsible for free annihilation of positrons, annihilations with electrons at vacancies, and annihilation of p-Ps. The component τ_2_ may result from the trapping of positrons in microvoids and in the clusters of free volumes of the amorphous state at the intersection of two or more grain boundaries. The third component, τ_3_, indicates the presence of free volumes, in which o-Ps are formed [11]. In the case of τ_1_ and τ_2_ annihilation, the so-called two-state model is applied to analyze the obtained results [2,6]. In positron annihilation from a free state and from one state located in the defect, in the absence of the detrapping process, a two-state model is used. Following this model, the numerical parameters of positron trapping can be calculated as follows:The mean positron lifetime τ_av_ characterizing the degree of deterioration of the environment prevailing in the tested glasses (the mean positron lifetime τ_av_ gives information about the electron density distribution in space)
(1)$$\tau_{\mathrm{av}\;}=\frac{\tau_{1}I_{1}+\tau_{2}I_{2}}{I_{1}+I_{2}}$$

The defect-free bulk positron lifetime τ_b_


(2)
$$\tau_{b}=\frac{I_{1}+I_{2}}{\frac{I_{1}}{\tau_{1}}+\frac{I_{2}}{\tau_{2}}}$$


The positron trapping rate κ_d_


(3)
$$\kappa_{d}=\frac{I_{2}}{I_{1}}\left(\frac{1}{\tau_{b}}+\frac{1}{\tau_{2}}\right)$$


The fraction of trapped positrons η


(4)
$$\eta =\tau_{1}\kappa_{d}$$


In 1960, Brandt, Berko, and Walker proposed the free-volume theory for positron annihilation. Initially, they assumed that the Ps was located in an interstitial region of the molecular network. They called this area free volume, defined as the difference between the cell and excluded volumes. Local free volumes appear due to the irregular packing of particles in the materials. Structural changes are combined with changes in the free volume [12,13,14]. The ratio between o-Ps life time and the free volume size is determined using the Tao–Eldrup model. This model assumes that o-Ps trapped in a spherical free volume can spontaneously annihilate with the emission of three γ quanta or as a consequence of the bounce process. In 1972, Tao and Eldrup developed this model for small voids such as void spaces in crystals, voids in polymers, or bubbles produced by the Ps in liquids [13,14]. Then, a simple model was studied, in which the Ps particle was placed in a spherical well with R_0_ radius and finite depth. To simplify the model’s calculations, the finite depth potential well was replaced with an infinitely deep spherical well but extended by ∆R. It was necessary to recreate the probability value of finding o-Ps outside the potential well, with a finite depth and R_0_ radius. The following theoretical considerations show that the lifetime τ o-Ps expressed as a function of the free volume of radius R is defined by the formula [14]:(5)$$\tau =0.5{\left[1-\frac{R}{R_{0}}+\frac{\sin2\pi \frac{R}{R_{0}}}{2\pi }\right]}^{-1}$$

In Equation (5), $R_{0}=R+\Delta R$, where ΔR is an empirical parameter. R was determined by matching the known sizes of holes and cavities in the molecular substrates. The best-defined ΔR value for all known data is 1.656 Å. Based on the model proposed by Tao and Eldrup (5), it is possible to link the lifetime of the ortho-positronium (through the “pick-off” process) τ_3_ with the size of the free volume penetrated by the posits in the material structure [14,15] as:(6)$$\tau_{3}\left[\mathrm{ns}\right]=0.5{\left[1-\frac{R}{R+\Delta R}+\frac{1}{2\pi }\sin\left(\frac{2\pi R}{R+\Delta R}\right)\right]}^{-1}$$where τ_3_ is the mean lifetime of o-Ps through the “pick-off” process expressed in ns and R is the mean radius of the free volume expressed in nm. In the above equation, after the empirical solution and determining R, we can calculate the free volume size $V_{f}$ according to the following formula:(7)$$V_{f}=\frac{4}{3}{\pi R}^{3}$$

The relative free volume, i.e., the ratio of the free volume to the macroscopic volume, is determined according to the semi-empirical formula [15]:(8)$$f_{v}={\mathrm{CV}}_{f}I_{3}$$where I_3_ is the intensity of the long-lived component in the positron lifetime spectrum expressed in [%]; C is an experimentally determined constant [10], which is 0.001–0.002; and V_f_ is the size of the free volume.

Vibrational spectroscopy, i.e., Raman spectroscopy, is a recognized and widely used method of structural research. The uniqueness and usefulness of this method for structural investigations results from its “sensitivity” to the subsequent ordering. It is beneficial for studies of amorphous materials with only medium and short-range ordering [16]. The application of Raman spectroscopy can also provide information on vibrations and rotation of bonds present in the glass matrix [16,17]. In addition, the spectral analysis allowed us to determine changes in the local structure due to doping with erbium ions.

The X-ray phase XRD analysis allows, among other things, to determine whether introducing an active dopant into the glass matrix of the tested material changes the ordering of the long-range structure [18].

Materials exhibiting magneto-optical properties are interesting due to the possibility of their use in optoelectronics, among others, such as magnetic field sensors or optical isolators used in laser systems for coupling semiconductor lasers with optical fibers to eliminate light reflected from the face of the optical fiber [19]. The main element of these optoelectronic devices is the Faraday rotators, whose operation is based on the Faraday magneto-optical effect, which consists of twisting the polarization plane of linearly polarized light passing through an isotropic transparent body under the influence of a magnetic field whose direction of the force lines is parallel to the direction of the light beam. Measurements of the Faraday effect allow one to determine the Verdet constant, which characterizes the magneto-optic properties of the tested material [20].

## 2. Results and Discussion

In this paper, six sets of tellurite glasses were investigated. Their concentrations are presented below:S1—70TeO_2_-5PbO-10P_2_O_5_-10ZnO-5PbF_2_;S2—70TeO_2_-5PbO-10P_2_O_5_-10ZnO-5PbF_2_+1200 [ppm] Er_2_O_3_;S3—70TeO_2_-5Bi_2_O_3_-10P_2_O_5_-10ZnO-5PbF_2_;S4—70TeO_2_-5Bi_2_O_3_-10P_2_O_5_-10ZnO-5PbF_2_+1200 [ppm] Er_2_O_3_;S5—70TeO_2_-5TiO_2_-10P_2_O_5_-10ZnO-5PbF_2_;S6—70TeO_2_-5TiO_2_-10P2O_5_-10ZnO-5PbF_2_+1200 [ppm] Er_2_O_3._

### 2.1. XRD

The X-ray diffraction scans measured for the tellurite glasses are shown in Figure 1. The obtained results confirmed the amorphous structure of all tested tellurite glass samples. Only wide convexities (amorphous halo) in the XRD images in 2θ range between 25 and 35 degrees, characteristic of the amorphous phase, are noticeable.

### 2.2. Positron Annihilation Lifetime Spectroscopy (PALS)

The analysis of the obtained lifetimes and their intensities for the investigated tellurite glasses doped with erbium ions Er^3+^ was carried out in two stages. First, the obtained results were considered using a two-state trapping model. In the following work, two-state and Tao–Eldrup models were used to determine the basic parameters describing the centers of positron annihilation in the investigated tellurite glasses. The research was a preliminary analysis of the internal structures of materials. Table 1 presents the results of the spectrum distribution positron lifetime for every sample. The obtained positron lifetime spectra showed that the incorporation of erbium oxide into the glass matrix leads to evident changes in the positron lifetime values τ_1_ and τ_2_ and their intensities I_1_ and I_2_ (Table 1). The fitting parameter of the PALS data, which is calculated by the LT9 is equal to χ = 1 ± 0.2. The calculations of the binary model and the Tao–Eldrup model are shown in Table 2. The κ_d_ and fraction of trapped positron η values are presented in Figure 2. Exemplary experimental PALS spectra measured for tellurite glass and tellurite glass doped with Er^3+^ions are shown in Figure 3.

After taking into account the analysis of the results obtained based on the two-state model (Table 2), it can be concluded:The doping of the basic glass with paramagnetic Er^3+^ions, practically, did not affect the mean positron life τ_av_ as well as their lifetime τ_b_ in the investigated samples.The κ_d_ for all tellurite glasses decreases during doping with Er^3+^ ions. The greatest decrease in this parameter was observed for sample S2, where this value decreased by 40%, and for samples S4 and S6, it decreased by 22% and 29%, respectively.The fraction of trapped positrons η, for all investigated tellurite glasses, also decreases during the doping with Er^3+^ ions.

Structural changes that occur as a result of doping the tested tellurite glasses gives us information that the doping of pure tellurite glass matrices with Er^3+^ ions affects the individual parameters of the two-state model. Thus, it enables further research aimed at doping tellurite glasses with more erbium and assessing the influence of the dopant on the glass properties.

In all the investigated tellurite glass systems, the third component τ_3_ with a value above 1 ns was obtained. This has not been achieved in studies conducted so far [3,19,21]. Obtaining the third component was probably possible by increasing the amount of the admixture (twice). The component τ_3_ is related to the annihilation associated with the “pick-off” process of trapping o-Ps through free volumes. All the free volumes in the test samples were not of the same size; therefore, the calculated values of the lifetime τ_3_ reflected the average sizes of the occurring free volumes. Since in this work, we were mainly interested in the changes of free volumes (sizes and their number) in the tested tellurite glasses doped with erbium ions, we considered two parameters obtained from the measurements, namely the lifetime values τ_3_ and its intensity I_3_. The o-Ps lifetime values τ_3_ and their intensities I_3_ are given in Table 1, while the sizes of free volumes R are given in Table 2. The obtained errors are the consequence of mathematical analysis.

### 2.3. Raman Spectroscopy

Figure 4 shows the normalized Raman spectra of the tested glasses. All spectra show a group of three bands in the range 400–550 cm^−1^ (bending vibrations of Te-O bonds), 600–900 cm^−1^ (stretching vibrations of Te-O bonds), and 940–1120 cm^−1^ (stretching vibrations of P-O bonds) [22,23,24,25,26,27,28,29].

As shown in Figure 5 and Figure 6, the shape of the S1–S4 spectra is almost identical. It indicates that the lead and bismuth ions play a similar role in bonding the tested glasses. It is known from the literature that the lead, bismuth, and niobium ions in the tellurite glasses play the role of modifiers [29].

As compared with the other spectra, a slightly different shape of the S5 and S6 spectra (the relation of the intensity of the bands in the range of 600–900 cm^−1^) indicates an additional role of titanium ions. However, due to the large half-width of the bands (amorphous materials), it is not easy to precisely interpret the spectra in question. Therefore, it was necessary to decompose them into component bands (Figure 5, Figure 6 and Figure 7, Table 3 and Table 4).

As it is clear to see, the decompositions of individual spectra show the same bands, the assignment of which to the respective vibrations is presented in Table 3 and Table 4. Based on the presented band assignments and the literature data, one can concluded that the incorporation of titanium ions in place of lead ions leads to a significant increase in the intensity of the D and C bands (Table 3 and Table 4), which is related to formation of the TeO_4_ units at the expense of TeO_3_ units (E and F bands). It indicates an increase in the degree of polymerization of the bond due to the formation of Te-O-Ti bridges. Thus, in the bond of the tested glasses, titanium ions play the role of network ions, and in this case, we are dealing with a tellurite–phosphate–titanium bond.

The comparison of the decomposition of individual spectral pairs (S1–S2, S3–S4, and S5–S6) also allows assessing the influence of erbium ions on their structure. The intensity of the integral individual bands (Table 3 and Table 4) shows that erbium ions depolymerize the bonding of the tested glasses. This is indicated by an increase in the intensity of the bands related to TeO_3_ units (E and F bands) at the expense of the intensity of the bands associated with TeO_4_ units (D and C bands). This effect is invisible with glasses containing titanium ions (S5–S6 pair), indicating that the titanium ions stabilize the tellurite subnetwork. The depolymerizing effect of erbium ions is also visible in the analysis of the phosphate subnetwork.

Based on the Raman spectrum (Figure 3, Figure 4 and Figure 5, Table 3 and Table 4), Q^0^, Q^1^, and Q^2^ units were found in all tested glasses (where Q is the number of bridged oxygen). The analysis of the corresponding glasses shows that the introduction of erbium ions leads to a decrease in the intensity of the bands associated with Q^2^ units. It indicates the depolymerization of the truss. Thus, it can be concluded that the erbium ions depolymerize only the phosphate subnetwork in S6 glass and simultaneously the tellurite and phosphate subnetwork in the case of S2 and S4 glasses.

### 2.4. Faraday Effect

Figure 8 shows the calculated maximum Verdet constants for the investigated tellurite glasses. The research of the Faraday effect allows one to observe the phenomenon of torsion of the plane of light polarization under the influence of the longitudinal magnetic field in each glass system.

The measurements of the Faraday effect confirmed the possibility of observing the phenomenon of torsion of the plane of light polarization under the influence of the longitudinal magnetic field in each glass system. The maximum error of the Verdet constant estimation was 4%. The greatest increase for the Verdet constant was observed for glasses where the modifier was titanium oxide. For the glass doped with erbium ions, the increase in the Verdet constant was more than 25% as compared with the glass containing bismuth oxide. Doping the glasses with paramagnetic erbium ions increases the angle of rotation of the plane of light polarization under the influence of the magnetic field in all tested glasses. Further research will be aimed at developing the optimal amount of erbium impurity in order to obtain the maximum value of the Verdet constant for the tested materials.

## 3. Materials and Methods

### 3.1. Materials

Melted 5 g sets of tellurite glasses (70TeO_2_-5XO-10P_2_O_5_-10ZnO-5PbF_2_, where X = Pb, Bi, Ti) were poured into a brass mold at a temperature of 330 °C. The samples were annealed at 320–340 °C for 2 h to remove any stresses developed in the material. Before starting the tests, the obtained glasses were carefully machined to form samples with geometrical parameters required for the measurement methods used. The nominal compositions of investigated glasses are summarized in Table 5. An example of tellurite glass after machining is shown in Figure 9. The sample is cylindrical in shape.

### 3.2. Positron Annihilation Lifetime Spectroscopy (PALS)

The sodium isotope ^22^Na is used as the source of positrons in PALS positron lifetime studies. The sodium isotope decays into neon ^22^Ne, emitting positrons e^+^ and gamma quanta of energy E_V_ = 1.2745 MeV. The registration of this quantity allows us to register the time of positron “birth.” In contrast, the time of “death” of the positron will be the registration of the gamma quantum resulting from the annihilation process inside the tested sample [11,12].

Defect parameters of the tellurite glass structure were investigated using positron annihilation lifetime spectroscopy (PALS). Positron lifetime measurements were performed using an Ortec start-stop spectrometer [11]. The spectrometer’s resolution FWHM was determined using the ^60^Co isotope, which was 250 ps. The ^22^Na isotope with an activity of 400 Bq, closed in Kapton foil, was used to measure the positron lifetimes. The source and the two samples formed a sandwich system during the measurements. As a result of PALS measurements, the positron lifetime spectra were obtained. To achieve repeatability of the results, the number of counts was equal to 2 × 10^6^. Subsequently, the spectra were analyzed using the LT9 program [30]. The LT program makes it possible to introduce a distribution of average lifetimes described by a function lognormal, whose fitting parameters are the mean value and the width of the distribution.

### 3.3. Raman Spectroscopy

The Raman spectra were obtained using a WITec confocal CRM alpha 300 Raman microscope equipped with an air-cooled solid-state laser, emitting wavelengths of 488 nm and a CCD detector cooled down to −82 °C. The power of the laser ranged from 14.4 to 14.6 mW, 120 or more scans, with integration time of 0.3–0.5 s and resolution equal to 3 cm^−1^, were collected and averaged.

### 3.4. XRD

X-ray diffractometry (XRD) was used to determine the phase constitution of the glasses. The XRD patterns were collected using a Bruker D8 Advance diffractometer with CuKα radiation equipped with a LynxEye detector. In order to obtain XRD data representative for the entire volume of the material, the samples were crushed to powder.

### 3.5. Faraday Effect

The magneto-optical properties of all investigated glass systems were determined based on the measurements of the Faraday effect as well as the calculations of the Verdet constant based on the value of the twist angle of the plane of light polarization. The maximum error in determining the Verdet constant was 4%. The measurements of the Faraday effect were carried out on a specially built measuring setup. Measurements of the angle of rotation of the polarization plane were made for a light wave ranging from 450 nm to 650 nm in a base magnetic field with an induction of 0.06 T. More information about the measurement method can be found in [31].

## 4. Conclusions

The XRD measurements confirmed the amorphous nature of the tested tellurite glasses.

Incorporation of the paramagnetic Er^3+^ ions into glass matrix caused changes in the parameters describing the structure of the glasses. We noticed that doping glasses with erbium ions clearly changed the positron trapping rate and the fraction of trapped positrons. Results of these measurements show that due to doping of tellurite glasses with erbium ions containing the optimal amount of erbium oxide (1200 ppm), it is possible to obtain the τ_3_ component and its I_3_ intensity, values related to the formation of ortho-positronium as well as the distribution of the density of the formation of free volumes. Notably, the research for the tellurite glasses presented in [21] could not obtain these values.

Raman spectroscopy reveal that the introduction of Er^3+^ ions into the structure of the tellurite glass breaks the Te-O-Te bridges and leads to the formation of new Te-O-Er bonds. The results of Faraday effect measurements indicate the possibility of changing the magneto-optical properties of tellurite glasses by the incorporation of various modifiers into the network and doping with rare earth metal ions.

Analyzing the obtained results of the conducted research, it was found that doping of tellurite glasses with erbium ions has significantly affects, with changes in the micro and nano structural properties.

## Figures and Tables

**Figure 1 ijms-24-03556-f001:**
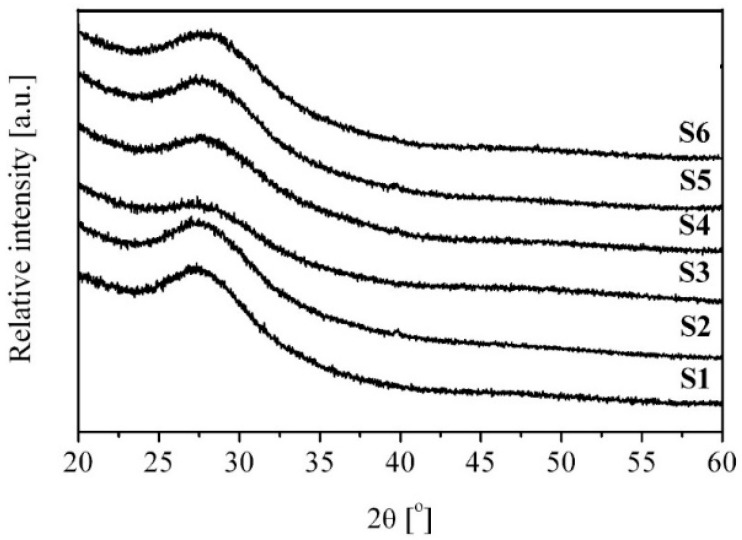
X-ray diffraction patterns of the investigated tellurite glasses.

**Figure 2 ijms-24-03556-f002:**
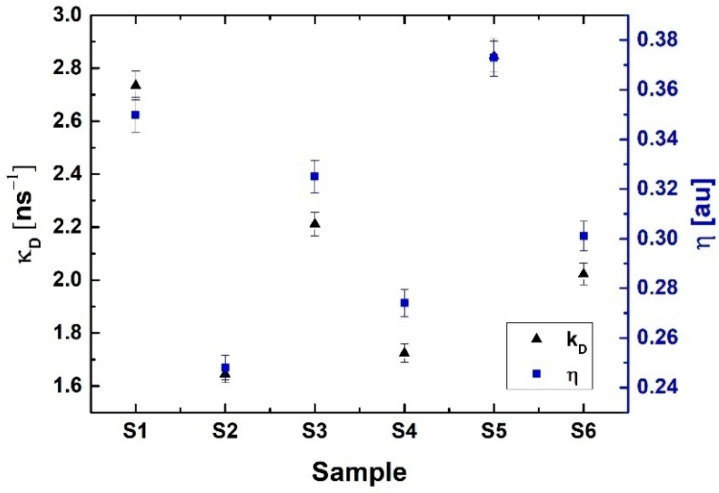
The positron trapping rate κ_d_ and the fraction of trapped positrons η calculated for all investigated samples.

**Figure 3 ijms-24-03556-f003:**
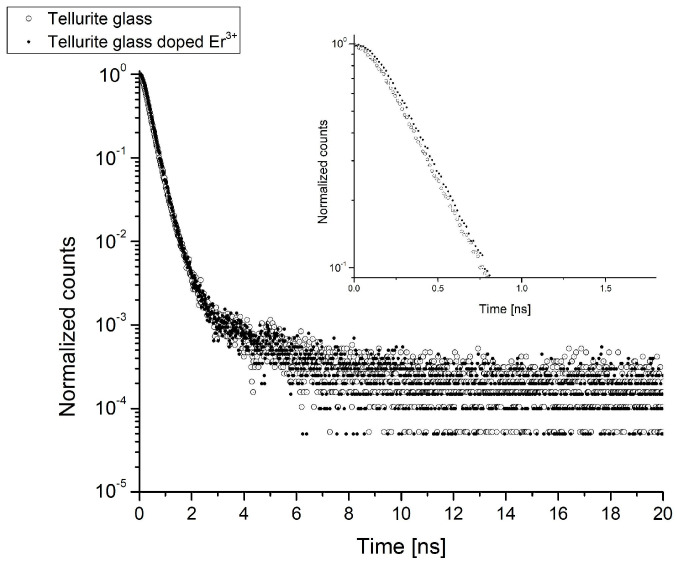
Exemplary experimental PALS spectra measured for tellurite glass and tellurite glass doped with Er^3+^ ions.

**Figure 4 ijms-24-03556-f004:**
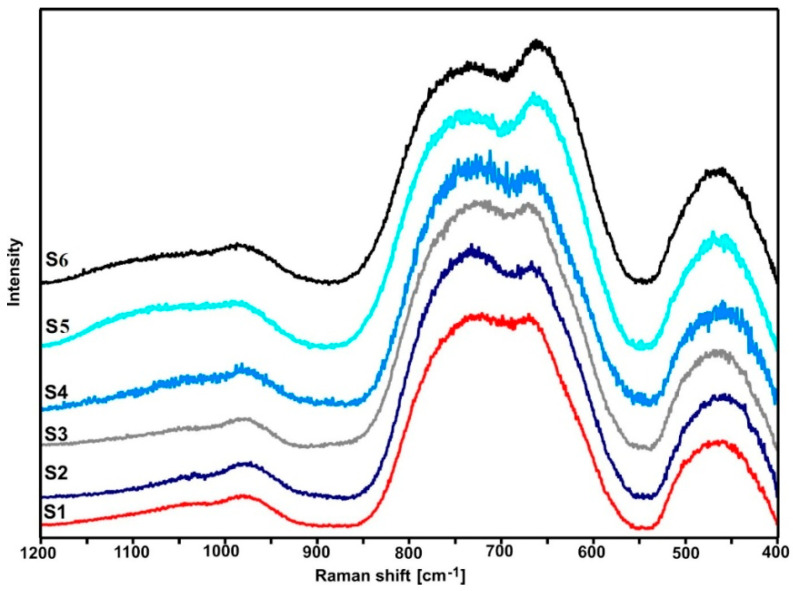
Normalized Raman spectra of S1–S6 glasses.

**Figure 5 ijms-24-03556-f005:**
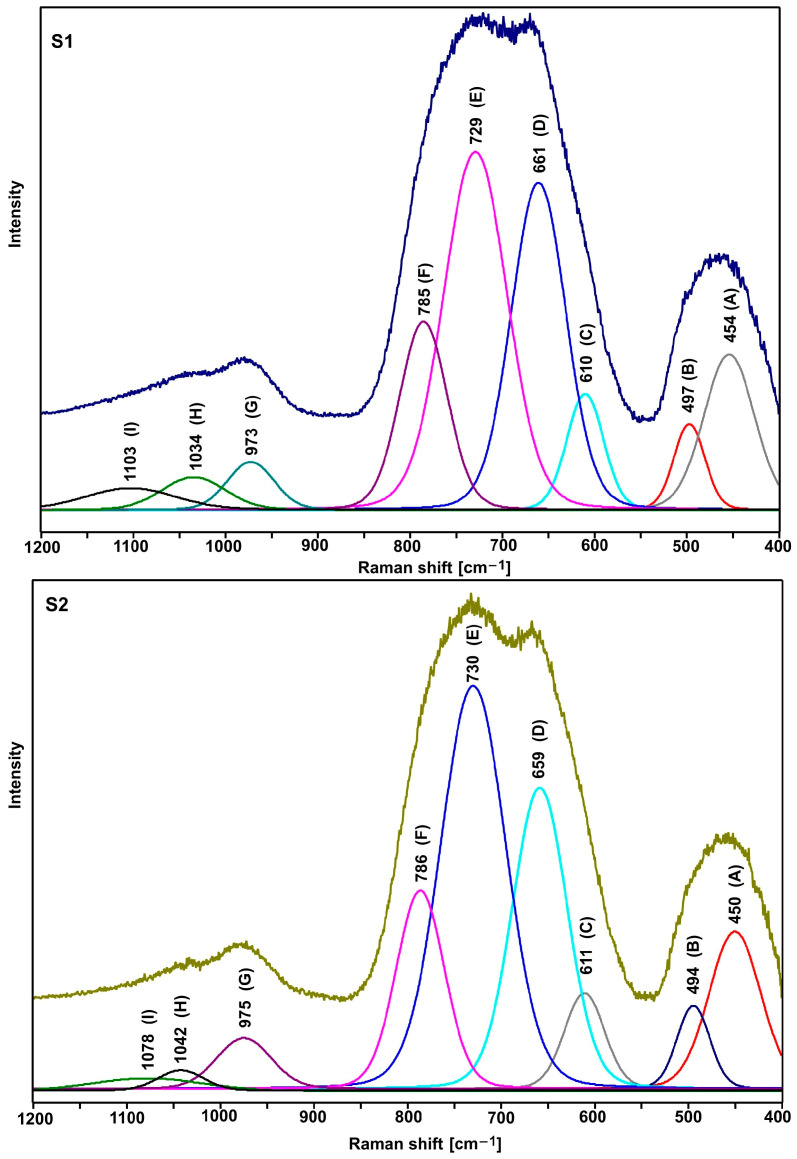
Deconvoluted Raman spectra of S1 and S2 glasses.

**Figure 6 ijms-24-03556-f006:**
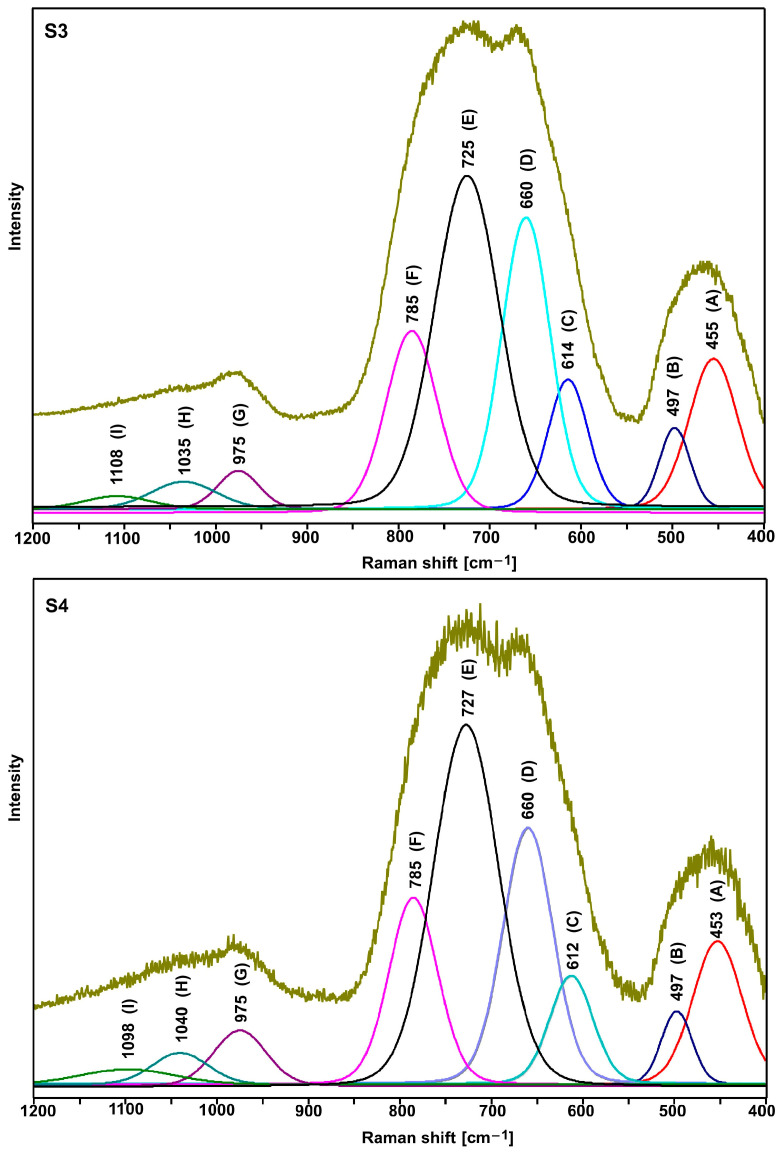
Deconvoluted spectra of S3 and S4 glasses.

**Figure 7 ijms-24-03556-f007:**
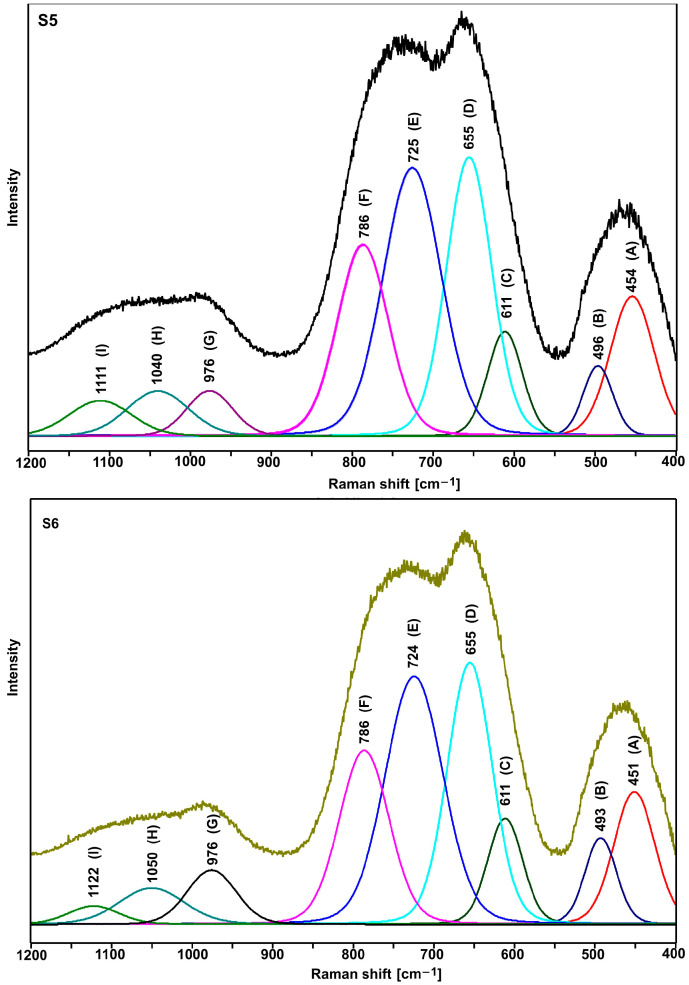
Deconvoluted Raman spectra of S5 and S6 glasses.

**Figure 8 ijms-24-03556-f008:**
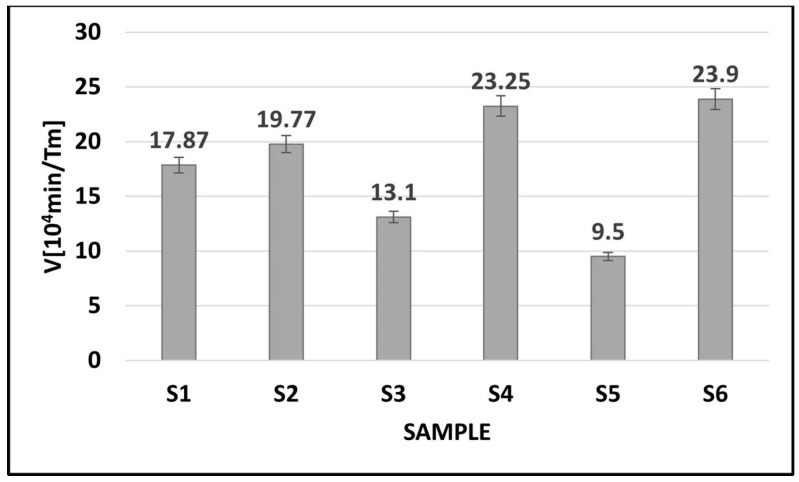
The maximum values of Verdet constants calculated for tellurite glasses.

**Figure 9 ijms-24-03556-f009:**
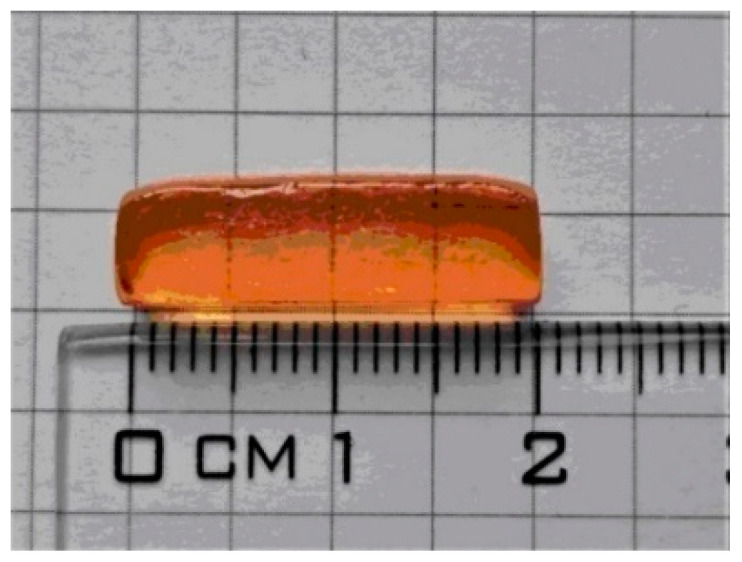
Tellurite glass after machining.

**Table 1 ijms-24-03556-t001:** Mean positron lifetime values of τ_1_, τ_2_, and τ_3_ and their intensities I_1_, I_2_, and I_3_.

SAMPLE	τ_1_ (ns)	I_1_ (%)	τ_2_ (ns)	I_2_ (%)	τ_3_ (ns)	I_3_ (%)
S1	0.128 ± 0.004	26.41 ± 1.27	0.245 ± 0.006	72.51 ± 2.04	1.920 ± 0.031	1.08 ± 0.05
S2	0.151 ± 0.006	38.95 ± 1.35	0.256 ± 0.007	59.81 ± 2.02	2.050 ± 0.040	1.04 ± 0.04
S3	0.147 ± 0.005	30.73 ± 1.32	0.278 ± 0.009	68.40 ± 1.76	1.941 ± 0.033	0.87 ± 0.05
S4	0.159 ± 0.007	32.31 ± 1.35	0.268 ± 0.009	66.81 ± 1.74	2.074 ± 0.034	0.88 ± 0.06
S5	0.131 ± 0.005	28.11 ± 1.31	0.273 ± 0.008	71.02 ± 2.05	1.834 ± 0.028	0.87 ± 0.06
S6	0.149 ± 0.006	33.91 ± 1.32	0.275 ± 0.008	65.20 ± 2.03	2.076 ± 0.029	0.89 ± 0.06

**Table 2 ijms-24-03556-t002:** Parameters of two-state trapping model and hole radius R sizes of free volume from the Tao–Eldrup model.

SAMPLE	τ_av_ (ns)	τ_b_ (ns)	κ_d_ (ns^−1^)	η (au)	R (nm)
S1	0.214	0.197	2.735	0.350	0.280
S2	0.215	0.201	1.645	0.248	0.292
S3	0.237	0.218	2.212	0.325	0.282
S4	0.228	0.208	1.724	0.274	0.294
S5	0.233	0.209	2.845	0.373	0.271
S6	0.232	0.213	2.023	0.301	0.295

**Table 3 ijms-24-03556-t003:** Component band’s parameters of Raman spectra of S1 and S2 glasses.

Band	Position (cm^−1^)		Integralintensity		Assignment
	TeO_2_-PbO	TeO_2_-PbO+Er_2_O_3_	TeO_2_-PbO	TeO_2_-PbO+Er_2_O_3_	[22,23,24,25,26,27,28,29]
A	454	450	55	56	Bending vibrations of O-Te-O bonds in TeO_4_ [28,29]
B	497	494	19	19	Bending vibrations of Te-O-Te bridge bonds in TeO_4_ units
C	610	611	25	29	Asymmetric stretching vibrations of the continuous network composed of TeO_4_
D	660	659	124	113	Asymmetric stretching vibrations of Te-O-Te bridge bonds occurring in TeO_4_ units that form a continuous network
E	729	730	157	163	Stretching vibrations of bonds between Te and non-bridging oxygens occurring in TeO_3+δ_ units [29]
F	785	786	62	65	Stretching vibration of TeO_3_ units
G	973	975	15	20	Symmetric stretching vibrations of non-bridging oxygens in Q^0^ units (PO_4_ tetrahedrons)
H	1034	1042	15	8	Stretching vibrations of P-O-P bonds in Q^1^ units
I	1103	1078	12	8	Symmetric stretching vibrations of non-bridging oxygens in Q^2^ units (PO_4_ tetrahedrons)

Q^n^-n is the number of bridging oxygens in phosphate (Q) units.

**Table 4 ijms-24-03556-t004:** Component band’s parameters of Raman spectra of S3–S4 pairs and S5–S6 pairs of glasses.

Band	Position (cm^−1^)		Integralintensity		Position (cm^−1^)		Integralintensity	
	TeO_2_-Bi_2_O_3_	TeO_2_-Bi_2_O_3_+Er_2_O_3_	TeO_2_-Bi_2_O_3_	TeO_2_-Bi_2_O_3_+Er_2_O_3_	TeO_2_-TiO_2_	TeO_2_-TiO_2_+Er_2_O_3_	TeO_2_-TiO_2_	TeO_2_-TiO_2_+Er_2_O_3_
A	455	453	53	51	454	451	57	54
B	497	497	18	17	496	493	19	27
C	614	612	37	36	611	611	34	37
D	660	660	101	95	655	655	117	117
E	725	727	160	165	725	724	143	142
F	785	785	70	75	786	786	90	86
G	975	975	11	19	976	976	19	21
H	1035	1040	13	11	1040	1050	26	25
I	1108	1098	10	8	1111	1122	20	9

**Table 5 ijms-24-03556-t005:** The chemical composition of the investigated tellurite glass systems.

Sample	Composition (%mol)
S1	70TeO_2_-5PbO-10P_2_O_5_-10ZnO-5PbF_2_
S2	70TeO_2_-5PbO-10P_2_O_5_-10ZnO-5PbF_2_+1200 [ppm] Er_2_O_3_
S3	70TeO_2_-5Bi_2_O_3_-10P_2_O_5_-10ZnO-5PbF_2_
S4	70TeO_2_-5Bi_2_O_3_-10P_2_O_5_-10ZnO-5PbF_2_+1200 [ppm] Er_2_O_3_
S5	70TeO_2_-5TiO_2_-10P_2_O_5_-10ZnO-5PbF_2_
S6	70TeO_2_-5TiO_2_-10P_2_O_5_-10ZnO-5PbF_2_+1200 [ppm] Er_2_O_3_

## Data Availability

Not applicable.

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
