# Peer review of "Tellurite Glasses from the 70TeO2-5XO-10P2O5-10ZnO-5PbF2(X= Pb, Bi, Ti) System Doped Erbium Ions—The Influence of Erbium on the Structure and Physical Properties"

_ijms, 2023, doi:10.3390/ijms24043556_

Round 1

Reviewer 1 Report

The paper report the influence of erbium doping on the structure and physical properties of tellurite glasses. The topic is interesting, however the manuscript needs a revision. English needs some polishing work and some points should be clarified. The manuscript is not suitable for publication in International Journal of Molecular Sciences in its present form, and need a major revision work. In the following my detailed comments and some general remarks:

General remarks

Sometimes the terminology is not appropriate, as examples Authors use “ionic rays” instead of “ionic radii” and several typos are present in the text. I suggest a careful revision of the text to avoid these inaccuracies. In my opinion, some information, which are given in the “material and methods” section, should be moved in the previous section. Authors label samples as S1-S6 in table 1 and 2 pag 5, but the definition of the labels is reported in table 5, pag 12. This is uncomfortable for the reader. Labels should be defined before their use, not 7 pages later.

Comments

-          Authors report in table 1 the positron lifetime t1 and t2 around 0.15 and 0.25 ns respectively. In the “material and methods” section they report that the spectrometer used for lifetime measurements has a resolution of 250 ps. Could Authors clarify this apparent incongruence?

-          The X scale of the inset in Fig 3 is not well suited, in my opinion.

-          Authors report, in tables 3 and 4, the position of the deconvoluted Raman spectra, labelled as A-I. In my opinion, Authors should label the peaks in the figures 5-7 to make easier the reading.

-          Line 207 “…. in Figure 5, the shape of the S1-S4 spectra ….” should be S1-S2 spectra or they should refer to figs 5 and 6.

-          Line 220 please check the sentence.

In summary, the work presents some novelties and could be considered for publication after a careful revision. I cannot recommend the publication on International Journal of Molecular Sciences in the present form.

Reviewer 2 Report

The work entitled “Tellurite glasses from the 70TeO2-5XO-10P2O5-10ZnO-5PbF2 (X 2 = Pb, Bi, Ti) system doped erbium ions - the influence of erbium 3 on the structure and physical properties” presents the results form a relevant work in the field. The manuscript is well written and is easy to follow in the description of the used methods and procedures. The conclusions are well supported by the results.

I only have some minor comments about it.

The introduction part regarding the PALS technique is well described, but in the way it is written it might be concluded as a general rule that 3 components are always present, and that o-Ps and p-Ps always present a significant contribution. It might be true in polymers or crystals, but not in metals or some semiconductors and other applications. I suggest changing a bit the description of PALS technique (lines 60 to 80) to clarify it.

Page 2, line 74. Authors wrote “mai result”, maybe you wanted to write instead “may result”?

No information about the fitting process of the PALS data is provided. I suggest including some comments about the fitting process regarding the used software (PALSfit maybe? Custom made software?) and the quality of the fit (that is indirectly reflected in the error bars of the lifetime components).

I suggest to include axis thicker line in Figure 8 (also in top and right sides of the plot to complete a “frame”) so the figure keeps the same aesthetics as the rest of the plots.
